# Mouthparts of the bumblebee (*Bombus terrestris*) exhibit poor acuity for the detection of pesticides in nectar

Rachel H Parkinson[1], Jennifer Scott[1], Anna L Dorling[1], Hannah Jones[2], Martha Haslam[1], Alex E McDermott-Roberts[1], Geraldine A Wright[1]*

[1]Department of Biology, University of Oxford, Oxford, United Kingdom; [2]Department of Life Sciences, Imperial College, London, United Kingdom

*For correspondence:
geraldine.wright@biology.ox.ac.uk

Competing interest: The authors declare that no competing interests exist.

**Abstract** Bees are important pollinators of agricultural crops, but their populations are at risk when pesticides are used. One of the largest risks bees face is poisoning of floral nectar and pollen by insecticides. Studies of bee detection of neonicotinoids have reported contradictory evidence about whether bees can taste these pesticides in sucrose solutions and hence avoid them. Here, we use an assay for the detection of food aversion combined with single-sensillum electrophysiology to test whether the mouthparts of the buff-tailed bumblebee (*Bombus terrestris*) detect the presence of pesticides in a solution that mimicked the nectar of oilseed rape (*Brassica napus*). Bees did not avoid consuming solutions containing concentrations of imidacloprid, thiamethoxam, clothianidin, or sulfoxaflor spanning six orders of magnitude, even when these solutions contained lethal doses. Only extremely high concentrations of the pesticides altered spiking in gustatory neurons through a slight reduction in firing rate or change in the rate of adaptation. These data provide strong evidence that bumblebees cannot detect or avoid field-relevant concentrations of pesticides using information from their mouthparts. As bees rarely contact floral nectar with other body parts, we predict that they are at high risk of unwittingly consuming pesticides in the nectar of pesticide-treated crops.

## eLife assessment

This study presents a **valuable** set of experiments to test whether *Bombus terrestris* bumble-bees can detect lethal-level doses of a series of pesticides in nectar-mimicking sugary solutions. Behavioural essays were coupled with electrophysiological measurements to show that *B. terrestris* mouthparts cannot detect high levels of the tested pesticides. If confirmed using pesticide formulas, and other bumblebee species, the study will be of general interest in environmental science research. Most experimental data are **compelling**, and the conclusions are sound, but the write-up would benefit from a broader ecological context.

## Introduction

Insect pollination is a critical ecosystem service. The most economically valuable pollinators are bees (*Calderone, 2012*; *Garibaldi et al., 2013*; *Huang and An, 2018*), especially domesticated bees that secure crop yield and quality (*Meehan et al., 2011*). In agricultural ecosystems, however, insecticides are widely used to protect crop yields (*Klein et al., 2007*). Using insecticides impacts the health and survival of bees, contributing to the recent population declines of all bee species (*Goulson et al., 2015*; *Potts et al., 2016*; *Van der Sluijs et al., 2013*; *Vanbergen et al., 2013*). Hundreds of studies have shown that exposure to sublethal concentrations of neonicotinoid pesticides (i.e. imidacloprid,

**eLife digest** Bees and other pollinators often encounter pesticides while collecting nectar and pollen from agricultural crops. Widely used to protect crops, pesticides are toxic to insects and have contributed to population declines in all bee species.

One way that bees might be able to avoid pesticides is using their incredibly good sense of taste, which can detect subtle differences between sugary solutions. Therefore, if pesticides taste bitter to them, bumblebees may be able to avoid feeding treated crops. However, it was not clear if bees can taste pesticides. Previous studies investigating whether they can taste a group of pesticides called "neonicotinoids" gave contradictory results. Furthermore, explicit behavioural tests of their ability to taste pesticides had not been performed.

To shed light on this, Parkinson et al. compared the responses of neurons within structures used for detecting taste in bumblees eating a pure sugar solution with those eating a solution containing pesticides. Experiments with a group of pesticides known as 'cholinergic' showed that neuron responses were the same whether the sugar solution contained pesticides or not. Secondly, by looking at bumblebee feeding behaviour, Parkinson et al. found that bees offered both pure and pesticide-laced sugar solutions would still drink the pesticide solution, even when it was toxic enough to make them very ill or kill them. This was the case regardless of which pesticide was used.

The experiments showed that bumblebees cannot use their sense of taste to avoid drinking pesticide-laced nectar, which is an important finding for policymakers making decisions about the use of pesticides on agricultural crops. It is possible that bees simply have a poor sense of bitter taste. However, in the future, these methods could be used to identify a compound that tastes bad to bees. Including such a compound in pesticides, could deter bees from feeding on pesticide-treated crops that do not require pollination, and help to restore their declining populations.

IMD, thiamethoxam, TMX, and clothianidin, CLO) in food impairs foraging behaviour, homing and orientation behaviour, and olfactory learning and memory (e.g. *Gill et al., 2012*; *Henry et al., 2012*; *Parkinson et al., 2022a*; *Schneider et al., 2012b*; *Williamson et al., 2014*; *Wright et al., 2015*). Most studies administer pesticides in sugar solutions; bees consume the solution, and then a change in behaviour or performance is recorded. If bees could detect and avoid pesticides in food, however, these substances could be used to defend crops against pests without risk to bees. Only a few have explicitly tested whether bees can taste and avoid neonicotinoid pesticides in nectar (*Arce et al., 2018*; *Kessler et al., 2015*; *Muth et al., 2020*), but the conclusions of these studies are contradictory. In addition, other compounds that affect cholinergic signalling in the insect nervous system, such as sulfoxaflor, have been proposed as alternatives to neonicotinoids, but whether or not these compounds can be detected by the bee's sense of taste has not previously been tested.

Insects detect non-volatile compounds (i.e. tastants) through activation of GRNs in contact chemosensilla on mouthparts, antennae, and tarsi which express receptor proteins that bind with specific types of ligands (e.g. sugars, non-nutrient 'bitter' compounds, salts, etc.; *Wright, 2016*). When a taste receptor binds with its ligand, this sets off a signal transduction cascade that results in depolarisation of the neuron (*Dethier, 1976*). For example, several types of receptors tuned to bitter substances are expressed in a specialized subset of GRNs (*Weiss et al., 2011*). When these GRNs spike in response to stimulation, feeding reflexes are inhibited (*French et al., 2015*). Bitter compound detection is also accomplished by the inactivation of sugar-sensing GRNs (*French et al., 2015*). For compounds which are potentially toxic, simultaneous activation of GRNs that directly inhibit feeding reflexes and silencing of GRNs that activate feeding make it possible to form a rapid response in reaction to contact (*Wright, 2016*). Thus, we expected that if bees could detect neonicotinoids as bitter compounds, we would see activation of bitter-sensing GRNs and/or inactivation of sugar-sensing GRNs. Previously, we observed that stimulation of the mouthpart's galeal sensilla with neonicotinoids in water did not elicit spikes from any GRNs, nor did we see a reduction in the rate of firing of sugar-sensing GRNs in the sensilla when stimulated with a mixture of sucrose and neonicotinoid compound (*Kessler et al., 2015*).

In a two choice assay, we unexpectedly found that when bees were given a choice between sucrose solution and sucrose containing field-relevant doses of IMD and TMX over a 24 hr period, the bees

choose the neonicotinoid solution (***Kessler et al., 2015***). This could indicate that bees find neonicotinoid pesticides phagostimulatory, like sugars. However, we also failed to find evidence that neonicotinoid pesticides in water elicited spikes in any galeal GRNs including the nutrient/sugar sensing neurons. For this reason, we concluded that bees could not taste neonicotinoids in nectar or on their own in water. This conclusion, however, has been challenged by a study in free-flying *B. terrestris* foragers (***Arce et al., 2018***). They found that free-flying bumblebees preferred specific concentrations of solutions laced with TMX, but only after a period of at least 10 days of feeding on the solutions (***Arce et al., 2018***). In this study, the location of the solutions was randomised, forcing a choice at the point of feeding. These authors concluded that the only way bees could solve this problem was if they used their sense of taste to identify the preferred solution, arguing that further work needed to be done to identify the mechanism for sensation. Subsequently, however, another study in a different species of bumblebees (*B. impatiens*) using methods similar to ours reported that bees are neither attracted nor deterred from consuming sugar solutions containing neonicotinoids, consistent with the idea that they cannot detect the pesticides (***Muth et al., 2020***). Thus, whether or not bumblebees can taste neonicotinoids in food remains unresolved.

Bees have highly specialized mechanisms for encoding sugars which are different to those reported for any other insect species. For example, in bumblebee galeal sensilla, stimulation with sugars produces coherent spiking in 2 GRNs which burst in response to stimulation with high-value saccharide compounds (***Miriyala et al., 2018***; ***Parkinson et al., 2022b***). We recently discovered that bumblebees also highly value the monosaccharide, fructose, and have a GRN tuned to detect it (***Parkinson et al., 2022b***). It is possible that changes to the burst pattern of firing could indicate that the presence of a compound such as a neonicotinoid in food. Furthermore, the type of sugar used to stimulate the sensilla could influence whether bees could taste neonicotinoids (***Parkinson et al., 2022b***). We also know that the labial palps are critical to toxin detection in other insect species (***Chapman and Sword, 1993***), and could house additional GRNs sensitive to bitter compounds like pesticides. Our previous work on neonicotinoid taste used sucrose to stimulate the galeal sensilla and did not analyse how pesticides might alter the burst structure of the galeal GRN response to sugars or test the labial palps. Clearly, deeper investigations could determine whether the mouthparts GRNs possess novel gustatory mechanisms for pesticide detection and provide critical information about the risk to bees of consuming the nectar of pesticide-defended crops in the field. They would also elucidate novel mechanisms for the detection of bitter compounds.

Here, we use a combination of sensitive behavioural assays and electrophysiology to test in detail whether bumblebee mouthparts have mechanisms to detect pesticides in nectar. Feeding assays using freely moving foragers make it possible to assess if bees detect and avoid potential toxins in food (***Ma et al., 2016***). To definitively test whether bumblebees detect neonicotinoids on their mouthparts, we quantified the structure of feeding when the mouthparts of freely-moving bumblebees (*B. terrestris*) were stimulated with nectar-like solutions containing pesticides over a 2 min period (***Ma et al., 2016***). We specifically tested a mixture of primarily fructose and glucose that mimicked the nectar of oilseed rape (*Brassica napus*) with field-relevant concentrations of neonicotinoids. In addition, we comprehensively tested these solutions on the gustatory receptor neurons in the A-type sensilla on the galea and the labial palps of the bees' mouthparts using electrophysiology to determine whether sensilla in multiple locations are able to detect neonicotinoids (IMD, CLO, and TMX) and the previously untested sulfoxamine pesticide, sulfoxaflor (SFX). The burst-spiking of the galeal GRNs was analysed for changes to structure and compared to the GRN response to the bitter compound, quinine (QUI), which has been reported previously to be a feeding deterrent to bees (***Ma et al., 2016***; ***Wright et al., 2010***).

## Results
### Sugar solution composition is encoded by GRNs
Responses of bumblebee galeal GRNs to individual sugar compounds have previously been described (***Miriyala et al., 2018***; ***Parkinson et al., 2022b***). Here, we tested whether a mixture of fructose, glucose, and negligible amounts of sucrose that mimics OSR nectar altered the GRN spiking responses compared to sucrose alone. We also expanded our investigation to include responses form GRNs in the labial palps, and tested A- and B-type sensilla on the labial palps and galea (***Figure 1A***). We found

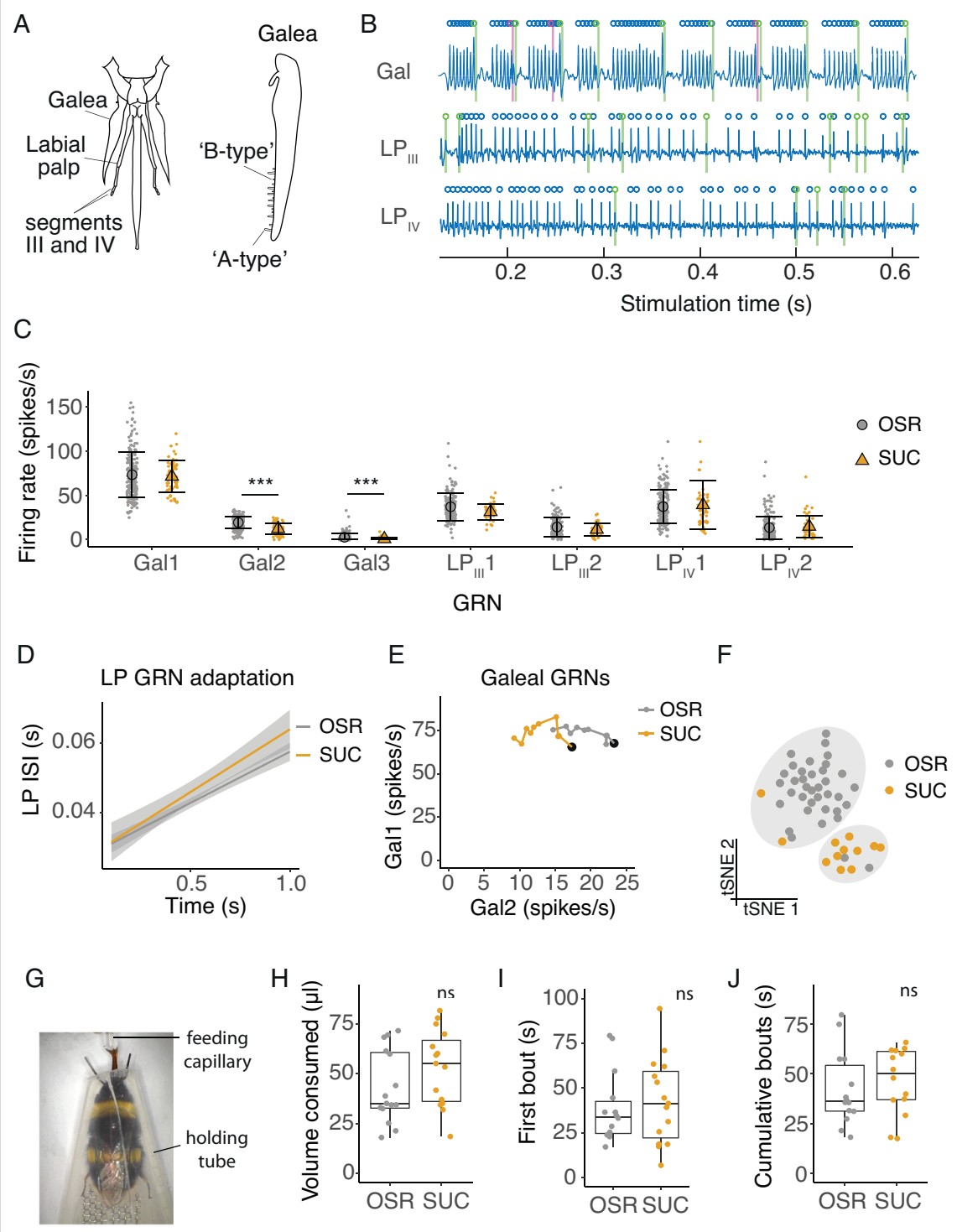

**Figure 1.** Electrophysiological and behavioural responses to sucrose and oilseed rape (OSR) nectar. (**A**) Diagram of the bumblebee's mouthparts from which tip-recordings were made, including the galea, and segments III and IV of the labial palps. Tip-recordings were made from the longer 'A-type' sensilla. (**B**) Filtered electrophysiological recordings from galeal (Gal), labial palp segment III (LP$_{III}$), and labial palp segment IV (LP$_{IV}$) sensilla. Spikes from GRN 1 at each location are labeled with blue circles, GRN 2 spikes in green, and GRN 3 spikes in magenta (only two GRNs present in labial palp recordings). (**C**) Average firing rates of GRNs from sensilla on the galea (Gal), segment III of the labial palps (LP$_{III}$), and segment IV (LP$_{IV}$) over 1 s of stimulation with 10% OSR or an equimolar (0.173 M) sucrose solution (SUC, n=315 sensilla from 37 bees). Mean and standard deviation illustrated with black symbols and bars, individual sensillum responses shown with coloured points. Asterisks represent significant differences between SUC and OSR (***, p < 0.0001). (**D**) Linear regression of inter-spike intervals per 0.1 s bin of labial palp GRNs versus time. Slope of regression is the adaptation rate

*Figure 1 continued on next page*

*Figure 1 continued*

and shading is standard error. There was no significant difference between the adaptation rate when stimulated with OSR versus SUC. (**E**) Firing rates of Gal1 versus Gal2 over 1 s stimulation with 10% OSR and SUC. Points represent mean rate in each 100 ms bin across all trials. A black marker highlights the first bin (i.e. time = 100ms). *Post hoc* comparisons showed that Gal2 firing rates were significantly different between stimuli. (**F**) t-SNE of all GRN responses for each animal following stimulation with OSR (gray) or SUC (orange) and k-means clusters (k=2, predicted by Monte Carlo reference-based consensus clustering) in gray shading. (**G**) Image of an untethered bumblebee in a holding tube feeding from a capillary in the 2-min feeding assay. (**H**) Total volume consumed of 1.79 M SUC or 100% OSR of freely moving bumblebees during 2 min (n = 15 bees per group). (**I**) The duration of the first feeding bout duration of bumblebees feeding on SUC or OSR. (**J**) The cumulative duration of feeding bouts within a 2-min period. For H-J, boxes show 25th, 50th and 75th percentile with 1.5x interquartile range (IQR) whiskers, with data from individual bees as coloured circles.

The online version of this article includes the following figure supplement(s) for figure 1:

**Figure supplement 1.** Electrophysiological responses to OSR.

**Figure supplement 2.** OSR and sucrose concentration gradients.

that GRNs in each location had specific patterns of activation that characterised their responses to sugar stimulation (*Figure 1B*). Stimulation of galeal sensilla with sucrose or with OSR nectar produced bursts of spikes resulting from the co-activation of two GRNs (Gal1 and Gal2), as we found for individual sugars previously (*Miriyala et al., 2018*; *Parkinson et al., 2022b*, *Figure 1B*). Each labial palp sensillum ($LP_{III}$ and $LP_{IV}$) contained two sugar-sensitive GRNs ($LP_{III}$ 1 and 2, $LP_{IV}$ 1 and 2), but these GRNs exhibited tonic firing patterns in response to stimulation instead of bursts of spikes (*Figure 1B*). The spiking rates averaged over 1 s stimulation of Gal1 and the labial palp GRNs were unchanged towards the two stimuli, but the average rate of Gal2 and Gal3 when stimulated with sucrose was lower than the rate for stimulation with 10% OSR (*Figure 1C*, LME, stimulus*GRN $F_{6,1607}$=9.16, *P*<0.0001). (Note: Gal3 was active only when stimulated with 10% OSR, but not by an equimolar sucrose solution, *Figure 1C*).

We compared the time-varying responses of GRNs using binned spikes (0.1 s bins over 1 s stimulation) averaged by animal across sensilla. The coefficient of variation of spike rates across sensilla for each animal for a given stimulus ranged from 0.023 to 2.44 with a median of 0.35 and IQR of 0.47. We found that the rate of adaptation, measured by the change in the interspike interval of labial palp GRNs, was not significantly different for 10% OSR or sucrose (*Figure 1D*, stim: $F_{1,461}$=3.11, p=0.079; time, $F_{1,450}$=586, p<0.0001; *Figure 1—figure supplement 1B*). However, galeal GRN burst structure (Gal1 and Gal2) differed between stimuli (*Figure 1E*, stim*GRN: $F_{1,938}$=13.8, p=0.0002, time: $F_{1,938}$=1.42, p=0.23). OSR elicited a 20% higher burst rate (i.e. Gal2 rate, *Figure 1—figure supplement 1C*) and shorter bursts (i.e. fewer Gal1 spikes per burst, *Figure 1—figure supplement 1D*), although the rate of adaptation of Gal1 did not differ as a function of stimulus (*Figure 1E*). To understand if the population of neurons we recorded from contained information about stimulus identity, we combined the temporal responses binned at 100ms intervals over 1 s of recording of all 7 GRNs across mouthparts using a clustering algorithm (t-SNE and k-means clustering). The algorithm predicted two statistically separate stimulation groups, one for OSR and one for sucrose (*Figure 1F*).

In spite of the differences observed in GRN responses, the total volume of the 10% OSR and equimolar sucrose solutions (1.79 M, *Figure 1G*) consumed by bumblebees was not significantly different in the free-feeding assay (*Figure 1H*, LME, $F_{1,28}$ = 2.09, p=0.16). We also did not observe a difference in the duration of the first feeding bout (*Figure 1I*, LME, $F_{1,26}$ = 0.176, p=0.678), or the total time the bees spent in contact with the solutions over 2 min (*Figure 1J*, LME, LME, $F_{1,26}$ = 0.555, p=0.463). We tested whether the concentration of these sugar solutions influenced behaviour using a concentration gradient of OSR and sucrose solutions at 2.0, 1.5, 1.0, 0.5 M and water. A greater volume of sucrose was consumed for the 1 M and 1.5 M concentrations than for the OSR solution, but there was no difference at any other concentration (*Figure 1—figure supplement 2*). First bout and total contact duration were not significantly different.

## Labial palp GRNs may enable bitter detection

QUI is routinely used to test an animal's sensitivity to bitter substances. Here, we tested a concentration series of QUI in 10% OSR solution to test how GRNs on the labial palps and galea respond to QUI, for use as a positive control for our tests with pesticides. A 1 mM QUI concentration suppressed all galeal and labial palp GRNs, while 0.1 mM QUI suppressed the activity of $LP_{III}2$ and $LP_{IV}2$ alone (*Figure 2A*, LME, stimulus*GRN $F_{12,1581}$=11.3, p<0.0001, *Figure 2—figure supplement 1A–C*). The

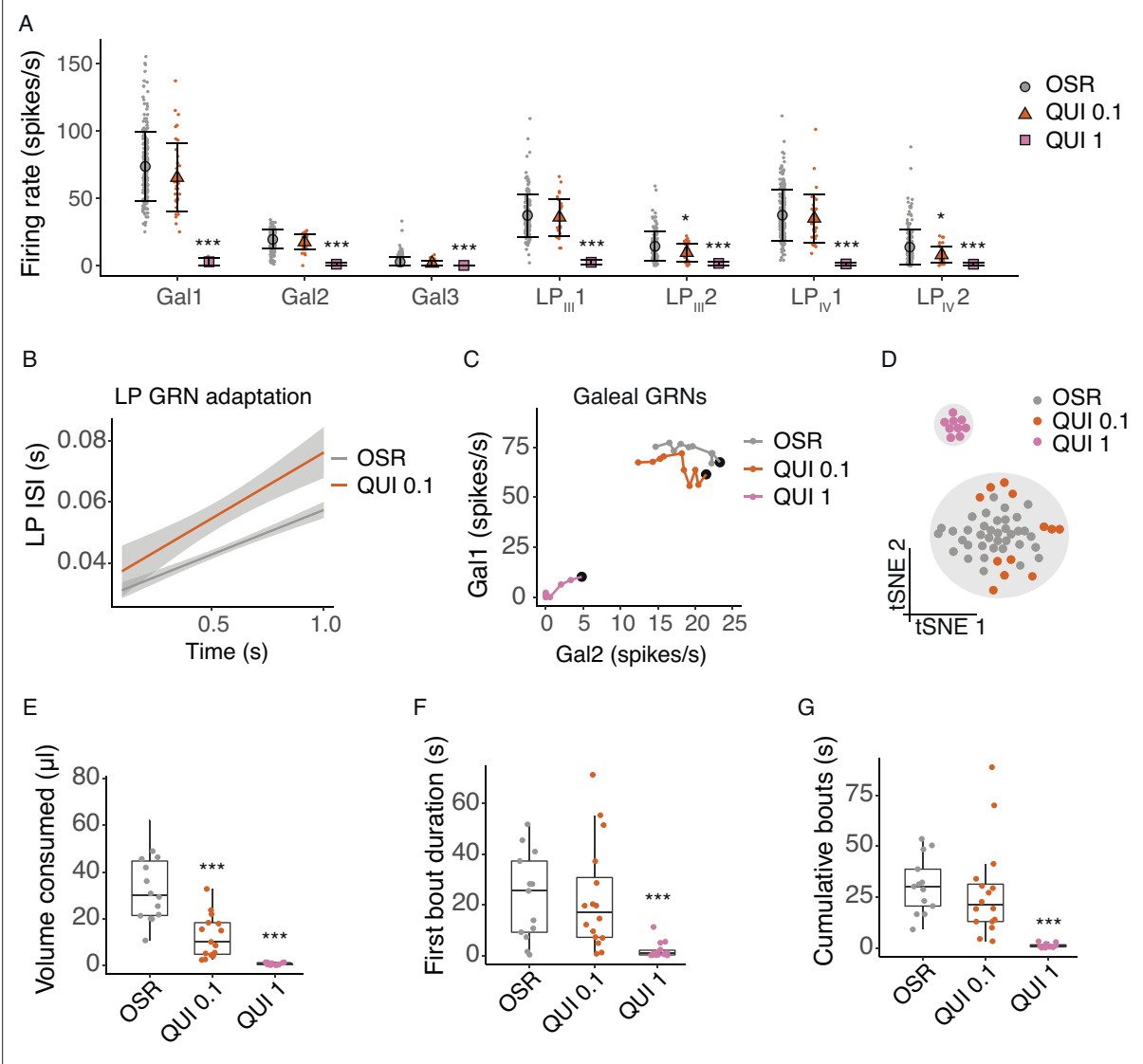

**Figure 2.** Quinine shuts down GRN and behavioural responses to sugars. (**A**) Average firing rates of 7 GRNs from the galea (Gal), and labial palps (LP_III, LP_IV) over a 1 s stimulations with 10% OSR (OSR), 10% OSR plus 0.1 mM quinine (QUI 0.1), and 10% OSR plus 1 mM quinine (QUI 1). Mean and standard deviation illustrated with black symbols and bars, responses of individual sensilla shown with coloured points (n = 55 sensilla from 17 bees). Asterisks represent significant differences between stimulus and OSR (*, <0.05; ** <0.001; ***, <0.0001). (**B**) Linear regressions of inter-spike intervals per 0.1 s bin of labial palp GRNs versus time illustrating the adaptation rate, with SEM in grey shading. The addition of 0.1 mM QUI to OSR significantly affected labial palp adaptation. 1 mM QUI not shown as there was insufficient spiking to calculate adaptation. (**C**) Firing rates of Gal1 versus Gal2 over 1 s stimulation. Points represent mean rate in each 100ms bin across all trials. A black marker highlights the first bin (i.e. time = 100 ms). *Post hoc* comparisons showed that only Gal1 (not Gal2) firing rates were significantly different at 0.1 mM QUI, while both were affected by 1.0 mM QUI. (**D**) t-SNE of all GRN responses for each animal following stimulation with 10% OSR, QUI 0.1 or QUI 1, and k-means clusters (k=2, predicted by Monte Carlo reference-based consensus clustering) in gray shading. (**E**) Volume consumed by freely-moving bumblebees of 100% OSR, OSR plus 0.1 mM QUI, or OSR plus 1 mM QUI over a 2 min period (n = 15 bees/group). Asterisks denote results of Dunnett's test (***, p<0.0001). (**F**) First bout duration when feeding on OSR, QUI 0.1 or QUI 1 (n=15 bees/group). Asterisks denote results of Dunnett's test (***, p<0.0001). (**G**) Cumulative bout duration over a 2 min period of bumblebees feeding on OSR, QUI 0.1 or QUI 1 (n=15 bees/group). Asterisks denote results of Dunnett's test (***, p<0.0001). For E-G, boxes show 25th, 50th and 75th percentile with 1.5x IQR whiskers, with data from individual bees as coloured circles.

The online version of this article includes the following figure supplement(s) for figure 2:

**Figure supplement 1.** Temporal responses of GRNs to quinine.

rate of adaptation, however, of the labial palp GRNs was faster when they were stimulated with the 0.1 mM QUI solution (*Figure 2B*, stim*time: $F_{1,450}$=13.8, p=0.0002). Galeal GRN burst structure was affected by the addition of QUI (*Figure 2C*, stim*GRN: $F_{2,1116}$=308, p<0.0001, time: $F_{1,1116}$=0.062, p=0.803; *Figure 2—figure supplement 1B–C*). Gal1 adapted faster when stimulated with 0.1 mM QUI compared to QUI, although there was no effect on Gal2 (*Figure 2C*). Clustering of all GRN responses predicted two groups, with the responses to 1 mM QUI separating from the responses to 10% OSR and 0.1 mM QUI (*Figure 2D*). OSR and 0.1 mM QUI were not significantly different.

The detection threshold observed in the behaviour experiments was lower than predicted by the t-SNE clustering but consistent with our measurements of adaptation in labial palp neurons. Bumblebees consumed significantly less OSR solution when it contained 0.1 mM QUI, while 1 mM QUI was completely deterrent (*Figure 2E*, LME, concentration: $F_{2,35}$=49.1, p<0.0001). However, the structure of feeding towards 1 mM QUI was similar to the results found for the cluster analysis of the GRNs: bees tested with 1 mM QUI exhibited much shorter average first feeding bout duration (*Figure 2F*, LME, concentration: $F_{2,40}$=18.2, p<0.0001) and shorter average cumulative bout duration (*Figure 2G*, LME, concentration: $F_{2,36}$=61.5, p<0.0001) than those fed with OSR or 0.1 mM QUI solution.

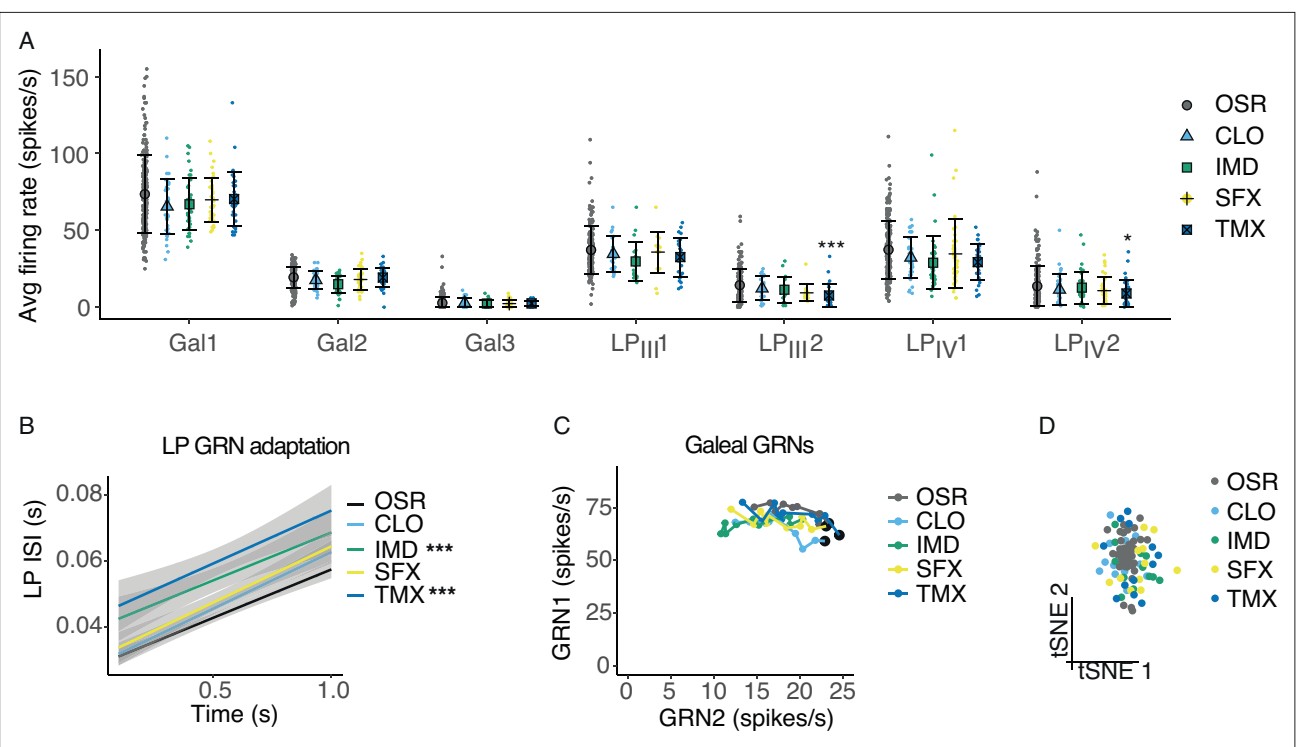

**Figure 3.** Pesticides affect the temporal code of bumblebee GRNs. (**A**) Average firing rates of 7 GRNs from the galea (Gal) and labial palps (LP$_{III}$, LP$_{IV}$) over a 1 s stimulation with 10% OSR (OSR), or OSR plus 0.1 mM clothianidin (CLO), imidacloprid (IMD), sulfoxaflor (SFX), or thiamethoxam (TMX, n = 331 sensilla from 37 bees). Mean and standard deviation illustrated with black bars, and data shown with coloured points. Asterisks represent significant differences between stimulus and OSR (*, <0.05; ** <0.001; ***, <0.0001). (**B**) Linear regressions of inter-spike intervals per 0.1s bin of labial palp GRNs versus time illustrating the adaptation rate, with SEM in grey shading. Asterisks represent significant differences between stimulus and OSR (*** p<0.0001). (**C**) Firing rates of Gal1 versus Gal2 over 1 s stimulation. Points represent mean rate in each 100 ms bin across all trials. A black marker highlights the first bin (i.e., time = 100 ms). *Post hoc* comparisons showed that only Gal1 firing rates were significantly different for CLO and IMD versus OSR, and Gal2 firing rates were affected by IMD only. (**D**) t-SNE of all GRN responses for each animal following stimulation with OSR, CLO, IMD, SFX, or TMX. Monte Carlo reference-based consensus clustering predicted an optimal cluster count of k=1 on the t-SNE.

The online version of this article includes the following figure supplement(s) for figure 3:

**Figure supplement 1.** Galeal and labial palp GRN cannot detect low concentrations of pesticides in nectar.

**Figure supplement 2.** Temporal responses of GRNs to high concentrations of pesticides.

## Extremely high concentrations of pesticides in OSR nectar reduce the rate of spiking of GRNs

In our previous work (*Kessler et al., 2015*), field-relevent concentrations of neonicotinoid pesticides were not detected by galeal GRNs. Here, we used OSR instead of sucrose and included sulfoxaflor. We also extended this experiment to test the labial palps. As before, we found no significant difference in galeal GRN responses between the control (OSR) solution and all of the solutions containing pesticides at field-relevant concentrations (*Figure 3—figure supplement 1*). No significant differences were detected in the responses of the labial palp neurons to the pesticide solutions (*Figure 3—figure supplement 1*).

To test whether all concentrations of pesticides go undetected by bees, we stimulated GRNs with 0.1 mM of each pesticide mixed with 10% OSR. Such concentrations are 10–100 k fold greater than has been reported for neonicotinoids from the nectar and pollen of agricultural crops and are at least 1 k fold greater than the $LD_{50}$ values for bumblebees (*Cresswell et al., 2014*; *Mundy-Heisz et al., 2022*; *Siviter et al., 2022*). Sulfoxaflor, however, is present at much higher concentrations in the field (e.g. 71.4 µM, *U.S. Environmental Protection Agency, 2014*). Stimulation with extremely high concentrations of TMX reduced the average firing rates of $LP_{III}2$ and $LP_{IV}2$; none of the other pesticides affected the average GRN firing rates (*Figure 3A*, LME, stimulus*GRN: $F_{24,2139}$=1.65, p=0.024.)

We also examined how the temporal response of galeal and labial palp neurons was influenced by stimulation with extreme concentrations of the pesticides. Labial palp GRNs adapted faster when IMD or TMX were added to 10% OSR (*Figure 3B*, stim: $F_{1,821}$=49.9, p<0.0001; time, $F_{1,808}$=571, p<0.0001), and firing rates of labial palp GRNs over 1 s were significantly reduced (*Figure 3—figure supplement 2A*). Gal1 and Gal2 spiking and adaptation rates were affected by CLO and IMD (*Figure 3C*, stim*GRN: $F_{4,1653}$=3.47, p=0.0078; time, $F_{1,1653}$=2.33, p=0.127). Galeal GRN bursting rate was slightly reduced when stimulated with OSR mixed with CLO or IMD (*Figure 3—figure supplement 2B*), while the burst length was not affected by the addition of pesticides (*Figure 3—figure supplement 2C*). We used the same clustering algorithm that was able to differentiate the responses to sucrose and OSR. Clustering using all of the data for each of the sensillum types failed to separate the pesticide responses from responses to the control (10% OSR, *Figure 3D*).

## Bumblebees readily consume pesticides in nectar

We assessed whether the pesticides were aversive in the range of values that included field-realistic concentrations (0–1000 nM) and one of the extreme values (0.1 mM) added to a 10% OSR solution. Field-relevant concentrations of the pesticides in OSR nectar did not deter the bees from feeding, as the presence of these compounds did not significantly alter the amount of food consumed (LME, concentration: $F_{4,297}$=0.209, p=0.933), first bout duration (LME, concentration: $F_{4,290}$=0.648, p=0.629) or cumulative bout duration (LME, concentration: $F_{4,288}$=0.310, p=0.871) over the 2 min assay period (*Figure 4A–C*). Importantly, we also tested feeding behaviour using an extreme concentration (0.1 mM) of the pesticides in OSR. Even at this very high concentration, we did not observe a significant difference in the amount of food consumed (LME: $F_{4,73}$=0.872, p=0.485), first bout duration (LME, $F_{4,74}$=0.392, p=0.814), or cumulative bout duration (LME, $F_{4,74}$=1.06, p=0.382) over 2 min (*Figure 4D–F*).

## Discussion

For the first time, we report how populations of GRNs on the bee's mouthparts respond to stimulation with putatively bitter compounds. We verified that bees do not have mechanisms on their mouthparts that enable them to taste neonicotinoids or sulfoxaflor when these compounds are present in nectar. Neonicotinoids and sulfoxaflor did not change the bursting of sugar-sensing neurons in the galea, nor did they elicit or inhibit spikes in labial palp neurons. Strikingly, we show that bumblebees do not avoid drinking OSR nectar solutions even when very high concentrations of common pesticides are present. Very high concentrations of these pesticides could produce a small but measurable reduction in the spiking of sugar-sensing GRNs, but these compounds never elicited spikes on their own. What's more, these small changes in the rate of spiking did not translate into features that the clustering algorithm could use to differentiate pesticide laced stimuli from the control OSR solution. From these data,

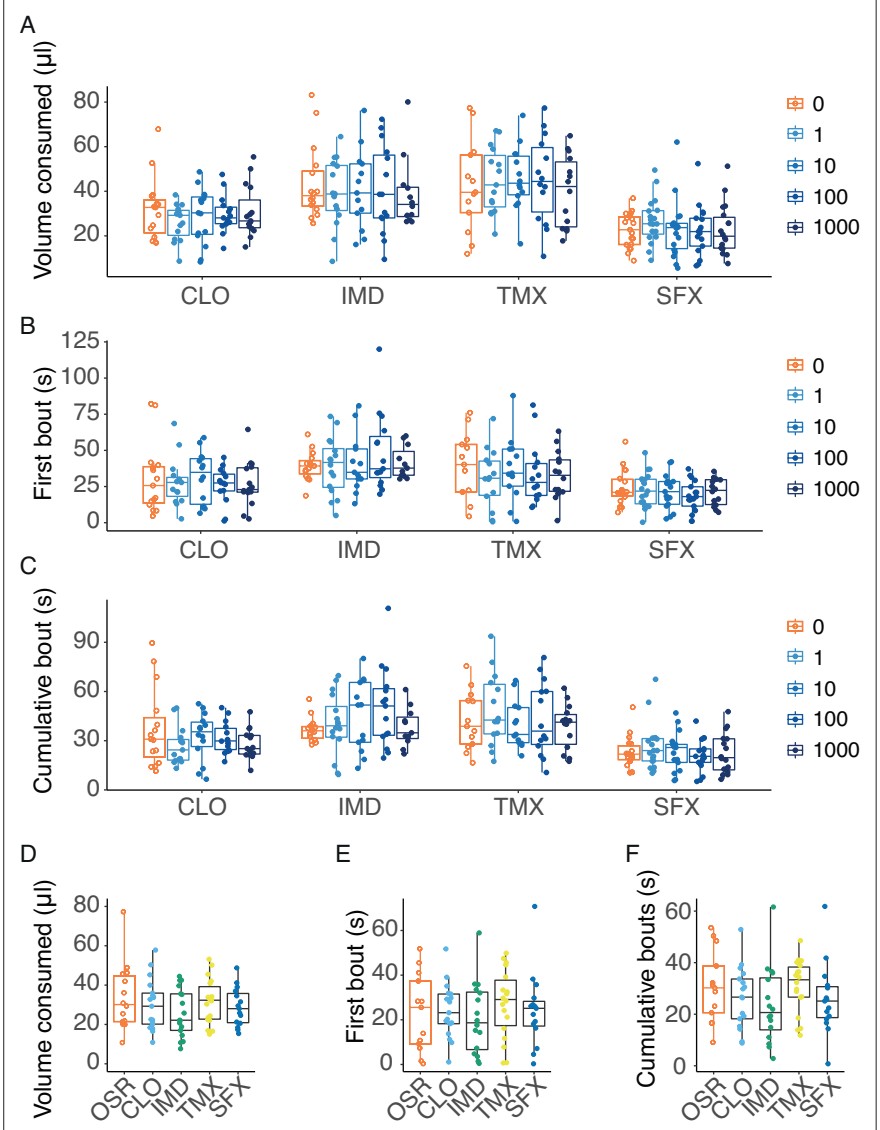

**Figure 4.** Bumblebees do not avoid consuming pesticides in nectar at field relevant concentrations. (**A**) The total volume consumed by freely moving bumblebees during 2 min of 100% OSR solutions containing increasing concentrations (nM) of CLO, IMD, TMX and SFX (n=15 bees per group). (**B**) First feeding bout duration of OSR mixtures with increasing concentrations (nM) of pesticides (CLO, IMD, SFX, TMX, n=15 bees per group). (**C**) Cumulative feeding duration during 2 min of OSR mixtures with increasing concentrations (nM) of pesticides (n=15 bees per group). (**D**) Total volume consumed of OSR or OSR plus 0.1 mM CLO, IMD, TMX or SFX over 2 min by freely moving bumblebees (n=15 bees per group). (**E**) First bout duration of bumblebees feeding on OSR and OSR mixed with 0.1 mM pesticides (n=15 bees per group). (**F**) Cumulative bout duration over 2 min by bees feeding on OSR and OSR mixed with 0.1 mM pesticides (n=15 bees per group). Boxes show 25th, 50th and 75th percentile with 1.5x IQR whiskers, with data from individual bees as coloured circles.

we conclude that the buff-tailed bumblebee's mouthparts do not have mechanisms for the detection and avoidance of common cholinergic pesticides in nectar.

In the present experiments, we expected that if neonicotinoids and sulfoxaflor could be tasted by bees, they would exhibit the same responses in GRNs produced by other non-nutrient compounds (i.e. bitter responses). For this reason, we were mainly testing for the reduction of GRN spiking caused by the suppression of sugar-sensing GRNs, as all of the 'bitter' compounds we tested were presented to the sensilla in OSR nectar (a mixture of primarily fructose and glucose). The known bitter compound, QUI, clearly suppressed the activity of sugar-sensing GRNs in both labial palps and the galea, but it

did not measurably alter the burst pattern of firing of the galeal GRNs. Importantly, our new data were able to confirm that field-relevant concentrations of pesticides did not significantly alter the pattern or rate of spiking of galeal or labial palp GRNs towards a nectar solution.

We used a clustering algorithm that integrated input from all the sensilla we recorded from over a 1 s time interval with every stimulus to identify whether population-level input made it possible to differentiate the stimuli. Although we were able to measure small but significant reduction in the responses of both labial palp and galeal neurons caused by the high concentrations (100 µM) of IMD, SFX, and TMX, this effect was not sufficient for the clustering algorithm to form classes of distinct stimuli for the OSR-pesticide mixtures. This is similar to what we observed for 0.1 mM QUI; although a small change in firing was detected, the clustering algorithm did not differentiate it from OSR. It is worth noting that the OSR background solutions in both cases were different: in the behavioural experiments the concentration of OSR was an average value of that naturally found in nectar. Our physiology experiments were performed using a 10% dilution of this solution. Different solutions were used because it was not possible to record from neurons with such a concentrated sugar solution, and in behavioural experiments, most bumblebees do not find 10% OSR sufficiently phagostimulatory to feed.

Our electrophysiological recordings are the first to report GRN responses from the bumblebee's labial palps. The labial palps in other insects such as locusts are used to detect both phagostimulants and deterrant compounds (*Blaney and Chapman, 1970*), so we might have rationally expected to observe labial palp GRNs that spike in response to QUI and cholinergic pesticides, but we did not. This could indicate that on the mouthparts, bitter compounds are only detected through the suppression of sugar-sensing GRNs (*de Brito Sanchez et al., 2005*).

It is interesting to note that in the case of all bitter compounds tested, the output of the clustering algorithm matched the behaviour of the bees, as the bees did not reject any of the pesticide solutions or the 0.1 mM QUI solution after the first feeding bout. This could indicate that that small differences detected at the sensory periphery do not translate into a sufficient signal across the whole population of gustatory neurons to produce an aversion to foods containing potential toxins. Bumblebees did drink less overall of the 0.1 mM QUI solution compared to OSR, however the amount of time they spent in contact with the feeding solution was the same. This suggests that, while 0.1 mM QUI is not immediately aversive to bumblebees (i.e. by taste), they can integrate negative post-ingestive feedback within a 2-min feeding period. Compounds reach the hemolymph of honeybees within 30 s of ingestion (*Simcock et al., 2018*), and quinine causes malaise-like symptoms (*Ayestaran et al., 2010*; *Hurst et al., 2014*), so it is feasible that the bumblebees in our behavioural assay factored in post-ingestive feedback while consuming 0.1 mM QUI. Unfortunately, when fed high concentrations of pesticides, post-ingestive feedback did not prevent bumblebees from consuming potentially lethal doses. For example, the LD50 for TMX in *Bombus terrestris* is approximately 6 ng/bee (*Siviter et al., 2022*), and bumblebees in our assay consumed a median amount of 3.5 ng of TMX when dissolved at 100 uM in OSR.

Our data also indicate that gustatory coding is more than the presence or absence of spikes arising from GRNs tuned to detect particular tastants. In bumblebees, it is clear that the pattern of spiking itself conveys information which is a feature that enables the brain to distinguish gustatory stimuli (*Parkinson et al., 2022b*). This supports the idea that encoding of gustatory stimuli is accomplished by the whole population of GRNs on a given body part (e.g. the proboscis) as was observed in multi-channel recordings from the adult hawkmoth maxillary nerve (*Reiter et al., 2015*). Thus, activity across the mouthparts GRN population that includes temporal input over time is likely to be used by the brain to differentiate taste stimuli and to facilitate the valuation of food quality. Our new data testing the difference between a nectar stimulus composed of a mixture of sugars (OSR) and sucrose clearly illustrates this. The changes in the rate of bursting and spiking in the galeal and labial palps were sufficient for the clustering algorithm to classify sucrose and OSR as separate stimuli. Although our behavioural assay did not show a difference, we would not expect it to, as it is an assay developed to study food rejection behaviour by bees, not differences in phagostimulation.

Previously, *Arce et al., 2018* reported that *B. terrestris* were capable of selectively foraging on 30% w/v sucrose solutions containing TMX. This finding confirmed our previous data that honeybees and bumblebees prefer solutions containing TMX (*Kessler et al., 2015*). *Arce et al., 2018* did not observe a preference for or against the solutions containing the neonicotinoids during the initial

days of their experiments. We expect that if taste was the mechanism for the preference, the bees would have shown a preference for the TMX solution within the first foraging bout, similar to the time frame of our behavioural assay in this study (2 min). Instead, the authors found that the preference for the TMX solution developed over time. Similarly, in our previous work, we found that individual bees developed a preference for solutions containing TMX if they were confined to feed on a choice between sucrose and sucrose laced with pesticide over longer time period (i.e. 24 hr, *Kessler et al., 2015*). We concluded that the preference was caused by the pharmacological action of TMX on cholinergic circuits involved in encoding reward in the insect brain (*Barnstedt et al., 2016*), as we found no electrophysiological evidence to support that bees could taste neonicotinoids (*Kessler et al., 2015*). This conclusion is consistent with the expectation that a preference resulting from post-ingestive reinforcement caused by amplification of the brain's response to sugars would be expected to develop following long-term, repeated exposure to a pharmacological compound (*Palmer et al., 2013*; *Wright et al., 2013*). If TMX caused bees to associate a higher 'reward value' with food found in a particular location, they would learn to return to it in preference over other solutions. Although *Arce et al., 2018* switched the feeders between training and testing phases, this may not have been sufficient to rule out the possibility that bees could learn the location of the food within a single feeding bout.

*Arce et al., 2018* conducted their experiments using gravity feeders in an enclosed arena, allowing bees to consume solutions ad libitum. Since the feeders did not have visual or olfactory cues, the authors inferred that the bees were capable of tasting the TMX in the solution. However, gravity feeders do not simulate how bees feed on real flowers. These feeders have large openings that allow multiple bees to feed simultaneously, enabling them to contact the solution with all body parts, including the tarsi and antennae. While it is possible that the sensilla on the antennae or tarsi could detect TMX, it seems unlikely given that honeybees have limited bitter detection in these locations, only observed when the sensilla are contacted with concentrations of compounds like QUI that are >1 mM (*de Brito Sanchez et al., 2005*; *de Brito Sanchez et al., 2014*). The field-relevant concentrations of the pesticides we tested are between 3–6 orders of magnitude lower in concentration (see *Kessler et al., 2015* for review); such dilute concentrations of a bitter compound present in a highly concentrated sugar solution would be difficult for most organisms to detect.

Floral morphology such as corolla length has often co-evolved with particular groups of pollinators with longer mouthparts; a long corolla protects nectar when visitors are efficient and reliable pollinators. In many flowers, nectar is hidden deep within the corolla such that the only contact that bumblebees have with it is via the distal end of the proboscis (also the location where most of the sensilla are). For example, in OSR flowers, nectaries are located at a depth of 5–8 mm (*Cresswell et al., 2001*) and the nectar cannot be contacted otherwise unless the bee bites through the petals. In addition, bees learn to handle flowers quickly to improve their foraging efficiency (*Heinrich, 1976*) and even extend the proboscis in anticipation of entering the corolla prior to landing. Thus, they are unlikely to contact nectar using gustatory sensilla on their antennae or tarsi while foraging. For this reason, it is likely that nectar palatability is largely determined by the sensilla on the proboscis.

## Materials and methods
### Bumblebee colonies
Bumblebee colonies (*Bombus terrestris audax,* Biobest, Westerlo, Belgium) were maintained at laboratory conditions (22–27°C and 35–40% RH) at the University of Oxford and fed ad libitum with the proprietary sugar syrup provided with the colonies (BioGluc, Biobest, Westerlo, Belgium) containing fructose (37.5%), glucose (34.5%), sucrose (25%), maltose (2%), oligosaccharides (1%), and the preservatives potassium sorbate (E202) 0.15% and citric acid (E330) 0.06% (*Wäckers et al., 2017*). Bumblebees were provided with freeze-dried honeybee collected pollen (approximately 10 g) (Agralan Growers, Wiltshire, UK) three times/week.

In total, 12 colonies were used for the taste assays and 6 colonies were used for the electrophysiological study. Females (of unknown age) were randomly assigned to a treatment and colonies were counterbalanced across treatments to control for intercolony variation.

## Solutions

An artificial oilseed rape nectar sugar solution was created containing 1.04 M glucose (D-(+)-Glucose, Sigma Aldrich), 0.746 M fructose (D-(-)-Fructose, Sigma Aldrich), and 0.007 M sucrose (Sigma Aldrich; *Carruthers et al., 2017*), hereafter referred to as 'OSR'.

Stock solutions (1 mM) were made for each pesticide in OSR solution. No solvents are needed at these concentrations as all pesticides used are readily soluble in water at concentrations far exceeding 1 mM. In order to assess field-relevant concentrations of each chemical, a concentration series of each of imidacloprid (IMD, Sigma Aldrich, CAS: 138261-41-3), thiamethoxam (TMX, Sigma Aldrich, CAS: 153719-23-4), clothianidin (CLO, Sigma Aldrich, CAS: 210880-92-5), and sulfoxaflor (SFX, Chem Service, CAS: 946578-00-3) was made via tenfold serial dilution using OSR to produce nectar containing the pesticides at 1000, 100, 10, and 1, and 0 nM (pure OSR). Concentrations of IMD, TMX, and CLO in floral nectar range from 1.3 to 64 nM (*Wood and Goulson, 2017*), and SFX can reach concentrations as high as 71.4 μM (U.S. Environmental Protection Agency (EPA), 2014). Additionally, 0.1 mM of each chemical dissolved in OSR was assessed alongside 0.1 mM and 1.0 mM QUI ((-)-Quinine hydrochloride dihydrate, Sigma Aldrich, CAS: 6119-47-7) in OSR, which functioned as positive controls for the effects of a bitter tastant. A concentration of 0.1 mM was chosen for the pesticides as this encompasses the higher range of SFX found in nectar, and allowed us to see if extremely high concentrations of IMD, TMX, and CLO could be detected by bees. All solutions were blinded.

## Aversive taste assay

Bumblebee workers were collected directly from the colony under red light and placed individually into plastic holding vials with ventilation holes drilled into the lid. To minimise inclusion of nest bees (i.e. bees that never or rarely forage), only workers with a thorax width >4.0 mm were used in these experiments (*Goulson et al., 2002*). In order to motivate the bees to feed during the assay, bees were deprived of food between 3 and 6 hr. Bees were held in individual plastic holding vials at laboratory conditions in total darkness throughout the starvation period. Following the starvation period, individual bees were transferred into a 15 ml falcon tube, modified such that the tip of the tube was removed and a small (10 mm x 20 mm) steel mesh was affixed contiguous with the resulting hole. Bees could extend their proboscis through this hole to reach the feeding solutions, whilst the mesh served as a surface for the bees to grip onto to maintain their position during feeding. The tube was then affixed to a polystyrene holder with dental wax (*Ma et al., 2016*).

A 60 mm long, 100 μL glass capillary tube was filled with a test solution and scanned onto a computer at 600 dpi to produce an image from which the start volume could be measured. The capillary tube was then connected to a silicone tube which was in turn connected to a 1.0 ml syringe via a female connector, as in *Ma et al., 2016*. This syringe functioned as a pipette bulb to maintain the feeding solution at the tip of the microcapillary. The capillary was held in place in the apparatus via a 1.0 ml modified syringe affixed to a micromanipulator. A DinoLite digital microscope camera (Model AM4815ZT, DinoLite, The Netherlands) was positioned 20 cm above the feeding site where the bees accessed the test solutions. This was connected to a computer and set to record using DinoLite Digital Microscope software (DinoLite, The Netherlands) at a frame rate of 640X480 at ×25 magnification. All feeding behaviour was recorded for later analysis.

Once a bee was positioned in the apparatus, they were given 3 min to habituate to their environment before the assay began. Following this, bees were encouraged to extend their proboscis by touching the bee's terminal antennomers with a droplet (~3.5 μL) of 0.5 M sucrose (Sigma Aldrich) dissolved in deionised water. Once the bee extended their proboscis, the droplet was presented to the bees' mouthparts for them to consume. Each bee was given up to 5 min to extend their proboscis and consume the droplet. Bees that did not do so were removed from the experiment. When the bee finished consuming the droplet, the microcapillary tube containing the test solution was presented to the bees extended mouthparts. The test phase began once the bee's proboscis contacted the test solution, and the bee was given 2 min to consume the solution. After feeding, microcapillaries were re-scanned to measure the end volume. A sample size of 15 bees per group was pre-determined using a power analysis and the results from previous studies (*Ma et al., 2016*).

All experimenters were blind to the experimental treatments. All solutions were tested in a randomised order and treatments were counterbalanced over time to eliminate any effects of starvation time on feeding behaviour.

## Measuring solution consumption and feeding behaviour

ImageJ (*Schneider et al., 2012a*) was used to produce measurements of the length of solution in each microcapillary before and after the assay. The reference scale was set to 60.0 mm. Image files were zoomed in to 400% and the length of the solution inside the microcapillary was measured meniscus to meniscus. The length of test solution consumed by each bee was calculated as the difference between the measured length of the liquid inside the microcapillary tube before and after the test phase. These lengths were then converted to volumes using the formula:

$$(100\mu L \times Xmm)/60.0mm$$

Where 100 µL is the maximum volume of the capillary tube, 60.0 mm is the length of the capillary tube, and X mm is the amount of solution consumed as a measured length within the capillary.

A feeding bout is defined as a period in which contact is made between the proboscis and test solution that is not separated by an absence of contacts for 5 s or more (*Ma et al., 2016*). The number of feeding bouts and their duration can be used to evaluate the phagostimulatory or deterrent activity of the compounds tested (*Ma et al., 2016*). To identify bouts of feeding behaviour, videos were played back at 50% speed and periods of contact between the bees' mouthparts and the solution were scored using the Noldus Observer (Noldus, the Netherlands). The duration of the first bout and the cumulative bout duration (over the 2-min period) were compared between stimuli.

## Electrophysiology

We performed extracellular tip recordings on individual taste sensilla on the bumblebee mouthparts to assess the peripheral taste perception of sucrose, a 10% dilution of oilseed rape nectar (10% OSR), and the addition of pesticides or QUI to 10% OSR. We recorded from 'A-type' sensilla from the galea (Gal), which are known to produce bursting spike trains in response to sugars (*Miriyala et al., 2018*) and the third and fourth segments of the labial palps (LP$_{III}$ and LP$_{IV}$, respectively, *Figure 1A*). Bumblebees were prepared for electrophysiology as described previously (*Miriyala et al., 2018*; *Parkinson et al., 2022b*). Briefly, bees were cold-anaesthetised and harnessed with the proboscis extended. The mouthpart nerves were severed by making an incision at the base of the mouthparts to prevent movement. The galea and labial palps were oriented for access to the sensilla using dental wax and wire pins. Sensilla were stimulated for 5 s with a borosilicate (Clark capillary glass 30–066, GC150TF-10) recording electrode (20 µm tip diameter, made with a Narishige PC-10 electrode puller) filled with the test solution. Signals were acquired with a pre-amplifier (TasteProbe; Syntech, Germany), amplified (AC amplifier 1800, A-M Systems, USA), digitised at 30 kHz (DT9803 Data Translation) and stored using DataView (version 11.5).

Pesticides (IMD, CLO, TMX, SFX) were tested in a background of 10% OSR. Pesticides were tested using a concentration gradient (1, 10, 100, and 1000 nM) and also at 0.1 mM. Additionally, QUI was tested in 10% OSR at concentrations of 0.1 and 1.0 mM. A minimum inter-stimulus interval of 3 min was used to prevent sensory adaptation. We obtained recordings from n=12 bee per compound and tested 2–4 sensilla per mouthpart location (6–12 sensilla per bee). To ensure the pesticides or QUI were not imposing a toxic effect on the taste sensilla, we included recordings at the beginning and end of each stimulation series with 10% OSR (*Figure 1—figure supplement 1*). Solutions were tested in a randomised order and treatments were counterbalanced over time to eliminate any effects of colony age on taste responses. Sample sizes were pre-determined with a power analysis and based on previous studies (*Parkinson et al., 2022b*). Sensilla were excluded from the analyses if they did not respond to the 10% OSR solution.

## Spike detection

Spikes from GRNs on the galea were extracted as described previously (*Parkinson et al., 2022b*). Briefly, after band-pass filtering (300–2500 Hz, second order butterworth filter) and normalisation, spikes were detected using the peakfinder function with adequate thresholds manually set for each recording. Interspike intervals were used to detect the end of burst positions (EOB). Using this method, we detected three GRNs within the galeal electrophysiological recordings, as seen previously (*Parkinson et al., 2022b*). GRNs of the labial palps were non-bursting, and visual inspection of the data suggested that two GRNs were present in these recordings. The spike waveforms for each recording were compared to differentiate the spikes from the putative units. Waveforms were

aligned from the peak +/-2ms, and the Matlab singular value decomposition (SVD) function was used to obtain the principal components of the spike waveforms. Waveforms were then projected onto the space spanned by the first two principal components, and k-means clustering was used to assign the waveforms to units for each recording. Using this method, two (or in some cases just one) GRNs were obtained from each recording, with the unit labelled GRN1 represented by larger waveform amplitudes and higher spike frequencies in response to sugars.

## Statistical analyses and data presentation

We performed all further analyses in R version 4.2.1 (*R Development Core Team, 2016*). Average firing rates of the 7 GRN types (Gal GRNs 1–3, LP$_{III}$ GRNs 1–2, and LP$_{IV}$ GRNs 1–2) across mouthpart locations were compared. Firing rates of a given GRN type were averaged across sensilla from a single animal, and log transformed ($\log_{10}$(average_rate +1)) to fit a normal distribution. We compared electrophysiological responses across stimuli using linear mixed effects models (LME, lmerTest package) (*Kuznetsova et al., 2017*) with Gaussian distributions and 'bee ID' as the random effect (avg_rate ~stimulus*GRN+(1|BeeID)). Models with non-significant interaction terms were re-run without the term (avg_rate ~stimulus + GRN+(1|BeeID)). Post hoc analyses were performed using estimated marginal means (emmeans package, *Lenth et al., 2018*) with Tukey's adjustment for family-wise error rates. Significant effects were denoted with asterisks (*,<0.05; **<0.001; ***,<0.0001).

We also assessed the temporal patterns of GRN activity over time. Time series spike data were averaged by animal for a given GRN type, and firing rates were calculated in 0.1 s bins from 0.1 to 1.1 s. Because the labial palp GRNs did not display a bursting pattern, we averaged the firing rate within each bin across all four labial palp GRNs. We assessed the bursting pattern of galeal GRNs (Gal 1 and 2) was assessed using two measures: the burst length (number of Gal1 spikes per burst) and burst rate. We constructed firing rate histograms (firing rate versus time) using these measurements over 1 s of stimulation, and compared stimuli with LME models (temporal_parameter ~stimulus + bin+(1|BeeID)) implemented in the lmerTest package (*Kuznetsova et al., 2017*). We denoted significant differences in the resulting slopes between stimuli on plots with letters, assessed with the estimated marginal means (emmeans) package (*Lenth et al., 2018*) with Tukey's adjustment for family-wise error rates. To assess differences in the complete temporal code of all 7 GRNs across mouthparts, we aligned the binned spike times in series for each GRN. For each animal and stimulus, there was a vector of 70 bins (ten 0.1 s bin per GRN type, aligned for 7 GRNs). To reduce the variability in responses between animals and balance the scale of responses across GRNs, we normalised the responses across GRN replicates within each animal by dividing by the average responses of each GRN to 10% OSR. The normalised responses of single GRNs were then averaged across replicates for each animal. We applied t-distributed stochastic neighbour embedding (t-SNE) to reduce the dimensionality of the temporal data into two components for each dataset (OSR vs sucrose, QUI, and pesticides), using Euclidean distance as the distance metric. The results from the t-SNE were subsequently clustered to reveal the grouping of responses by stimulus. To predict the optimal cluster count, we employed Monte Carlo reference-based consensus clustering (M3C, *John et al., 2020*), an unsupervised learning technique that aggregates across multiple k-means clustering runs. If the optimal number of clusters (k) was greater than one, we included clustering on the t-SNE plots.

Measurements of feeding behaviour (volume consumed, first bout, and cumulative bout) were compared using t-tests LME, with 'colony' as a random effect: (feeding_measurement ~stimulus + (1|ColonyID)), or (feeding_measurement ~stimulus + concentration+(1|ColonyID)) (lmerTest package) (*Kuznetsova et al., 2017*), with emmeans *post hoc* tests (emmeans package) (*Lenth et al., 2018*) and Tukey's adjustment for family-wise error rates. The non-significant interaction term (stimulus*concentration) was removed for all pesticide concentration gradient models due to increased model fit (using AIC and BIC). Behavioural data that was not normally distributed was log transformed. For all boxplots, the median is indicated by a line, bounds of the box mark the 1st and 3rd quartile, and whiskers extend to 1.5 times the interquartile range (IQR). Where data points are drawn, these represent the responses of individual GRNs.

## Acknowledgements

We thank Priya Chakrabarti-Basu for the pilot studies that led to this work. Funding: This research was supported by a BBSRC grant (BB/S000402/1) to GA Wright, and an NSERC PDF (PDF-546125–2020) and Royal Society Newton International Fellowship (NIF/R1/19368) to RH Parkinson.

## Additional information

### Funding

| Funder | Grant reference number | Author |
|---|---|---|
| Biotechnology and Biological Sciences Research Council | BB/S000402/1 | Geraldine A Wright |
| Royal Society | NIF/R1/19368 | Rachel H Parkinson |
| Natural Sciences and Engineering Research Council of Canada | PDF-546125-2020 | Rachel H Parkinson |

The funders had no role in study design, data collection and interpretation, or the decision to submit the work for publication.

### Author contributions

Rachel H Parkinson, Conceptualization, Data curation, Formal analysis, Funding acquisition, Visualization, Writing – original draft, Writing – review and editing; Jennifer Scott, Data curation, Writing – review and editing; Anna L Dorling, Data curation, Writing – original draft; Hannah Jones, Martha Haslam, Alex E McDermott-Roberts, Data curation; Geraldine A Wright, Conceptualization, Supervision, Funding acquisition, Methodology, Writing – original draft, Project administration, Writing – review and editing

### Author ORCIDs

Rachel H Parkinson http://orcid.org/0000-0002-8192-3178
Anna L Dorling https://orcid.org/0009-0005-4084-0201
Hannah Jones http://orcid.org/0000-0002-9481-8094
Geraldine A Wright https://orcid.org/0000-0002-2749-021X

Reviewer #1 (Public Review): https://doi.org/10.7554/eLife.89129.3.sa1
Reviewer #2 (Public Review): https://doi.org/10.7554/eLife.89129.3.sa2
Author Response https://doi.org/10.7554/eLife.89129.3.sa3

## Additional files

### Supplementary files

• MDAR checklist

### Data availability

Data in this manuscript are currently available on Figshare.com at https://doi.org/10.6084/m9.figshare.22005272.v1.

The following dataset was generated:

| Author(s) | Year | Dataset title | Dataset URL | Database and Identifier |
|---|---|---|---|---|
| Wright G | 2023 | Mouthparts of the bumblebee (Bombus terrestris) exhibit poor acuity for the detection of pesticides in nectar | https://doi.org/10.6084/m9.figshare.22005272.v1 | figshare, 10.6084/m9.figshare.22005272.v1 |

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
