## [Editor Report · eLife assessment]

This study presents a **valuable** set of experiments to test whether *Bombus terrestris* bumblebees can detect lethal-level doses of a series of pesticides in nectar-mimicking sugary solutions. Behavioural essays were coupled with electrophysiological measurements to show that *B. terrestris* mouthparts cannot detect high levels of the tested pesticides. If confirmed using pesticide formulas, and other bumblebee species, the study will be of general interest in environmental science research. Most experimental data are **compelling**, and the conclusions are sound, but the write-up would benefit from a broader ecological context.

---

## [Referee Report · Reviewer #1 (Public Review)]

Summary:

Parkinson and colleagues address an interesting and important question, i.e. whether the bumblebee Bombus terrestris can receive field-realistic concentrations of different pesticides in a sugar solution mimicking nectar. The study directly follows up on a previous study conducted by the same team (Kessler et al. 2015, Nature), which was partly questioned by another more recent study (Arce et al. 2018, Proc R. Soc. B). The authors apply a combination of electrophysiological measurements and behavioral feeding tests to answer this question. Their results strongly suggest that B. terrestris workers are not able to perceive field-realistic doses of pesticides in a sugar solution. They additionally show that B. terrestris can physiologically differentiate between solutions varying in sugar composition.

Strengths:

Sophisticated methodology, combination of approaches, clear and precise language. The stats questions have been addressed to my satisfaction. In terms of interpretation, however, several suggestions and comments were provided from an ecological perspective, which was deemed important, while the authors have expressed their intent to concentrate on the electrophysiological mechanism. Given that this study was motivated by conflicting results from earlier research, which were frequently employed to discuss the authors' findings, I still find that the discussion needs to be expanded in order to encompass a wider context.

---

## [Referee Report · Reviewer #2 (Public Review)]

Summary:

This manuscript is part of the Wright lab's ongoing studies that investigate whether the bumblebee B. terrestris can detect the presence of pesticides when feeding. Previously, they showed that B. terrestris cannot detect neonicotinoids and would prefer food containing neonicotinoids (Kessler et al. 2015). However, in that paper, they showed that B. terrestris cannot taste neonicotinoids but did not provide evidence on why B. terrestris prefer food containing neonicotinoids. In the current paper, the authors continue to suggest that B. terrestris cannot taste neonicotinoids as well as another insecticide, sulfoxaflor, based on additional behavioral experiments and electrophysiological experiments focusing on specific GRNs. While the data from these experiments continue to suggest that B. terrestris cannot taste these insecticides using their mouthparts, whether B. terrestris can actually perceive these insecticides, and why this species prefers food containing these compounds remains unknown.

Strengths:

The authors provided additional evidence that B. terrestris cannot taste neonicotinoids with their mouthparts. The authors have addressed my concerns regarding overgeneralization and that parts of the manuscript were written in a way that sounded combative with studies from other groups that had come to slightly different conclusions from their previous paper.

---

## [Author Response]

The following is the authors’ response to the original reviews.

Response to Public Reviews

Reviewer #1:

We thank this reviewer for their comments on our paper. We have adjusted the methods section to ensure it is clear, including an updated description of the stastics and in some cases updated stastical methods to ensure uniformity in analyses across datasets. The discussion has been modified so that the message regarding our results is set appropriately in the literature.

Reviewer #2:

We are grateful to this reviewer for their insight. We have modified the text of the discussion to address the points of this reviewer, including providing a greater focus on the significance of our results without overgeneralizing. We have addionally reframed our argument regarding the detection of pescides by *Bombus terrestris* to more carefully consider conflicing results from other papers.

Response to Recommendaons For The Authors

Response to Reviewer #1

We thank this reviewer for their thoughtful comments and ideas. We have made several changes to the text of the manuscript to improve the clarity of our writing, and we are grateful to the reviewer for raising several important points that we had not sufficiently discussed in the paper previously. We feel the paper has been improved with the inclusion of a more thorough discussion and clarified methods. Please see below our responses to the points they raised.

A few general thoughts that I had when reading your manuscript: I assume you have only tested the acve pescide ingredients, but not the formula generally applied in the field. Given that these formulas contain addional compounds but the acve ingredients, might it not be possible that they could be perceived by bees?

For this study, we were interested specifically with the taste of active pesticide compounds, although we agree it could be interesting to explore the taste of other formula compounds, it was not within the scope of this paper to test these.

Is there an alternave to quinine as a negave control? As you state, quinine is generally used in studies and likely oen in concentraons which are beyond what can be seen in e.g. floral nectar, which might explain its aversive effect. I would like to see it tested in natural concentraons and ideally in combinaon with other potenally toxic plant secondary metabolites (PSMs).

The purpose of including quinine in our study was to provide an in-depth characterizaon of “bitter” taste responses using the sensilla on bumblebee labial palps and galea (i.e., through the atenuation of GRN firing). This was not to show how bumblebees may interact with plants containing quinine in the field, or other PSMs. It would indeed be interesting to explore other plant secondary metabolites, however this was beyond the scope of our paper.

L177-187 AND 233-238 Could you, please, provide a photo or schemac drawing to illustrate your assay?I have a very hard me picturing the actual set-up.

We have provided a labeled diagram of the bumblebee mouthparts and sensillum types (Fig 1A), as well as an image of the bumblebee feeding from a capillary in the behavioural assay (Fig 1G). Further details about the feeding assay (including a JoVe video) can be found with the Ma 2016 paper that we cite throughout our methods section.

L219 Why did you choose 5 sec here?

This feeding bout duration was selected based on the criteria defined in Ma et al 2016. We have added a citation to that sentence.

L221-224 How precisely was the behavior scored? Just length of bouts or also repeated short contacts? Please, specify.

We used the first bout duration and the cumulative bout duration in our analyses. A sentence has been added to specify this.

L231/233 Please, provide some brief details here, as many readers may find it annoying to find and read another study for methods' details.

We have added three sentences in the methods to further explain the electrophysiological method.

L238-245 See also my general methods comment: concentraons used for pescides and quinine differ quite substanally, which may explain their different effects on the bees' percepon. Are the concentraons used for quinine realisc? If not that is totally fine for a negave control, but it would be interesng to see a comparison of effects conducted for similar concentraons.

The concentrations used of quinine were selected so that they would elicit a known “biter response” – these concentrations are not meant to be field-realistic, and our data (and others, e.g., Tiedeken et al 2014) show that lower concentrations of quinine are not detected by bumblebees.

L277-301 I assume that this is a standard stascal approach to analyze electrophysiological data. However, I am really struggling with fully understanding what you did here. It might be good to add some addional explanaon/detail, e.g. on why you constructed firing rate histograms or how you derived slopes (aren't smulus and bin categorical variables?).

Firing rate histograms are indeed very commonly used for visualizing neuron spikes over me. We have changed the text somewhat in an effort to make it more clear. Likewise, the “slopes” were derived from the LMEs, and in this case “bin” is a connuous me variable – any me variable will involve some binning depending on the resolution used but should not be considered categorical.

L291-295 As you were averaging and normalizing your data, could you, please, provide some informaon on variaon within animals?

We have now included information on the coefficient of variation for spike rates across sensilla for a given animal/smulus (CV range, median, and IQR).

L295 I assume t-SNE represent a mulvariate approach for ordinaon, correct? Can you explain why you chose to use this approach? Did you use Euclidean Distance?

Yes, t-SNE is a mulvariate technique for dimensionality reduction. It is parcularly well-suited for the visualizaon of high-dimensional datasets, as it can reveal the underlying structure of the data by embedding it in a lower-dimensional space (e.g., 2D) while preserving the local structure of the data as much as possible. We used t-SNE because it has been shown to be effecve in visualizing clusters of similar data points in high-dimensional data. Euclidean distance was used as the distance metric for the t-SNE embedding. Euclidean distance is the default distance metric for most implementaons of t-SNE and is appropriate for connuous data like the spike counts in this study. We have adjusted the methods to clarify this.

L304 Why did you not always use LMEs?

We have adjusted the text to show that we used LME for all comparisons, and the statistics have been updated accordingly in the results secon. None of the outcomes changed with the implementation of LME for all tests.

L306 Would it not make sense to also include the interacon between smulus and concentraon in your models?

We have now included a sentence to explain that the interacon term was removed due to it being nonsignificant, and the models without the interacon term having beter model fit (determined by having lower AIC and BIC values).

Results:L337, 339 and more: I would prefer to see actual p-values, not just "p > 0.05".

This has been adjusted on L337 and 339. As far as we are aware, there are no other instances where exact p-values were not given (except when p < 0.0001).

Discussion:L470 This is true, but the bees' behavior changed significantly, indicang that they may respond to this small change in firing paterns already?

It is true that the bees’ behaviour changed significantly with 0.1mM QUI, but this was not the case with the pescides. Bees drank less overall of 0.1mM QUI than OSR because of the rapid posngesve effects of this compound. It’s important that the duraon of the first bout was not affected (i.e., they didn’t directly avoid it by taste upon first encountering it, as they do with 1mM QUI), but just that they drank less of the 0.1mM QUI over 2 minutes. Post-ingesve effects may occur as quickly as 30s aer inial consumpon. For the pescides, the small changes in GRN firing were not associated with any effects on consumpon or our other measures of feeding behaviour, and we suggest this results from a lack of rapid negave posngesve consequences. We now include further discussion of these important points.

L481 But they consumed significantly less of the 0.1 mM QUI!?

This is true, but they did not reject it (i.e., not drink it at all), and there were no changes in the amount of me the bees spent in contact with the 0.1mM QUI soluon compared to OSR. We have adjusted the text for clarificaon.

L504/505 AND 520/521 AND 533-536 I see your point, but I am wondering whether the bees may need some me but are generally able to learn the taste of pescides, which may explain why e.g. Arce et al. only saw an effect over me. For example, learning to 'focus' on the pescide taste may require some internal feedback (bees not feeling well) or larvae feedback. If I understood right, you tested workers only, which might be less sensive than queens or larvae. I think these aspects should be discussed.

In our study, we invesgated the mechanism of taste detecon of pescides. We agree that bees likely use posngesve mechanisms to learn to associate the locaon (or another cue) of a food source with posive or negave posngesve cues. ‘Focus’ is a higher-order process that involves increased atenon to sensory smuli but does not affect sensaon at the level of the receptor. We show that bees are unable to taste pescides using the gustatory receptors on their mouthparts, so post-ingesve learning would not be able to associate the pescides with any taste cue. Indeed, there may be caste-specific differences with foraging queens, however a discussion of this would be beyond the scope of our paper.

I also recommend broadening the scope of your discussion. For example, you only focus on nectar, while the story might be different for pollen, which is also contaminated with pescides but represents a different chemical matrix with potenally different taste properes. Also, unlike nectar, pollen is collected with tarsae and hence on contact with other bee body parts.I would also like to see a discussion of your study's implicaons for other bee species and other potenally toxic compounds (e.g. PSMs).

We do not include any data in this paper regarding tarsal or antennal taste or other potenally toxic compounds. In this paper we present one mechanism of biter taste percepon (i.e., of quinine) and specifically show that the buff-tailed bumblebee is unable to taste certain pescides using their mouthparts. To avoid overgeneralizing, we have not included discussions about other species or compounds, which may or may not share similaries with our study.

Response to Reviewer #2

We thank this reviewer for their comments. We have adjusted the text to avoid overgeneralizaons with our conclusions, and atempted to soen language in the discussion that may have been construed as combave towards the Arce et al (2018) paper. We hope this reviewer finds these adjustments to be in line with their expectaons.

1. In two parts of the manuscript, the authors made broad predicons and conclusions beyond what the evidence in the paper can support and wrote "Future studies will be necessary to confirm this." (Lines 508-509) and " Future studies that survey a greater variety of compounds will be necessary to confirm this." (563-564). Instead of making conclusions based on what experimental data in future studies may support, I would ask the authors instead to make conclusions that their current study can support based on experimental evidence in this paper.

We have removed these predicons that extend beyond the scope of the paper.

1. Line 315 "GRNs encode differences in sugar soluon composion". The logic of the tle is wrong.

This has been fixed.

1. Since this study is only performed in one bumblebee species, then I would suggest that the tle be more specific e.g., "Mouthparts of the bumblebee Bombus terrestris exhibit poor acuity for the detecon of pescides in nectar".

We have made this change.